

# Variant profiling of evolving prokaryotic populations

Markus Zojer[1], Lisa N. Schuster[2], Frederik Schulz[3], Alexander Pfundner[1], Matthias Horn[2] and Thomas Rattei[1]

[1] Department of Microbiology and Ecosystems Science, Division of Computational Systems Biology, University of Vienna, Vienna, Austria
[2] Department of Microbiology and Ecosystems Science, Division of Microbial Ecology, University of Vienna, Vienna, Austria
[3] DOE Joint Genome Institute, Lawrence Berkeley National Lab, Walnut Creek, CA, United States

## ABSTRACT

Genomic heterogeneity of bacterial species is observed and studied in experimental evolution experiments and clinical diagnostics, and occurs as micro-diversity of natural habitats. The challenge for genome research is to accurately capture this heterogeneity with the currently used short sequencing reads. Recent advances in NGS technologies improved the speed and coverage and thus allowed for deep sequencing of bacterial populations. This facilitates the quantitative assessment of genomic heterogeneity, including low frequency alleles or haplotypes. However, false positive variant predictions due to sequencing errors and mapping artifacts of short reads need to be prevented. We therefore created VarCap, a workflow for the reliable prediction of different types of variants even at low frequencies. In order to predict SNPs, InDels and structural variations, we evaluated the sensitivity and accuracy of different software tools using synthetic read data. The results suggested that the best sensitivity could be reached by a union of different tools, however at the price of increased false positives. We identified possible reasons for false predictions and used this knowledge to improve the accuracy by post-filtering the predicted variants according to properties such as frequency, coverage, genomic environment/localization and co-localization with other variants. We observed that best precision was achieved by using an intersection of at least two tools per variant. This resulted in the reliable prediction of variants above a minimum relative abundance of 2%. VarCap is designed for being routinely used within experimental evolution experiments or for clinical diagnostics. The detected variants are reported as frequencies within a VCF file and as a graphical overview of the distribution of the different variant/allele/haplotype frequencies. The source code of VarCap is available at https://github.com/ma2o/VarCap. In order to provide this workflow to a broad community, we implemented VarCap on a Galaxy webserver, which is accessible at http://galaxy.csb.univie.ac.at.

Corresponding author
Thomas Rattei,
thomas.rattei@univie.ac.at

## INTRODUCTION

The genotyping of heterogeneous populations of one prokaryotic species is an increasingly important method to address microbiological questions regarding population composition

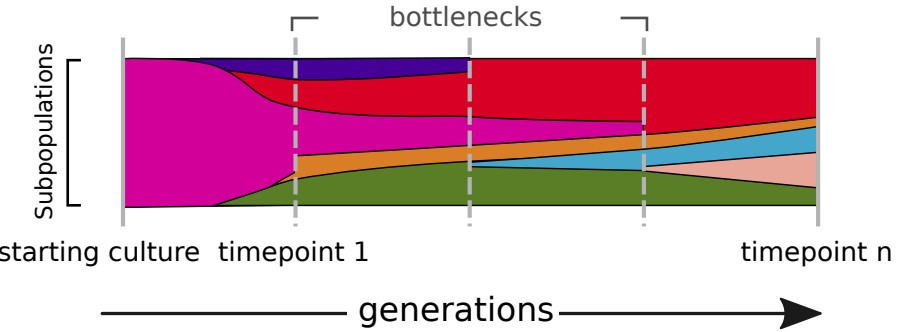

**Figure 1** **The dynamics of a bacterial population.** Alternate alleles arise over successive generations. They encounter positive or negative selection and bottlenecks, which shape the diversity landscape of a population over time.

and dynamics under prevalent selective pressures. This approach is, for example, used in experimental evolution (EE) experiments (*Barrick & Lenski, 2013*) and studies of host—pathogen systems (*Gardy et al., 2011*; *Bos et al., 2011*; *McElroy, Thomas & Luciani, 2014*). Recent developments in Next-Generation-Sequencing (NGS) technologies allow for sequencing at high coverage within a short timeframe, however limited to short read length.

The classical approach of assembling genomes out of short DNA reads preferably reconstructs the most abundant genotype into genome contigs and scaffolds. In order to retrieve haplotype frequency information, reads need to be mapped onto the assembly or a reference genome. Variant calling is then performed on the alignment of the reads. The predicted variants can be phased into haplotypes or alleles if a whole haplotype reconstruction is not possible due to insufficient linkage of the variant sites. The variant prediction, however, can lead to false positives due to sequencing errors, such as InDels and substitutions. The reads may be misplaced during mapping due to their short length and thus can lead to false positive variant calls (*Li, 2014*). Sequencing errors can be partially reduced by quality filtering and error correction (*Yang, Chockalingam & Aluru, 2013*). As a consequence, the substitution error rate for Illumina could be decreased below one percent while InDel homopolymer errors showed to accumulate logarithmically with the length of the stretches (*Minoche, Dohm & Himmelbauer, 2011*) and can thereby be reliably identified.

In evolving populations, we expect a heterogeneous mix of variant alleles (Fig. 1). Most of the genotyping studies of prokaryotes so far have been done by resequencing of clonal bacterial cultures (*Maharjan et al., 2013*; *Blount et al., 2012*). The technique of deep sequencing of non-clonal populations, named Pool-seq, was mainly done for metagenomic profiling of communities (*Qin et al., 2010*) and only to a minor extend for the characterization of allele frequencies (*Eyre et al., 2013*; *Khan et al., 2011*; *Köser et al., 2012*; *Pulido-Tamayo et al., 2015*). The genotyping of non-clonal variants in heterogeneous populations, however, remains challenging (*DePristo et al., 2011*; *Nielsen et al., 2011*; *Kofler & Schlötterer, 2014*; *Pulido-Tamayo et al., 2015*).

In order to get a most complete picture of the different haplotype or allele frequencies, it is fundamental to use Pool-seq and exploit high coverage sequencing data to detect all types of variants, which are SNPs, InDels and structural variations (SV). One way to deal with

this is to integrate several variant calling software tools, which utilize different approaches for the detection of the different kinds of variants.

Commonly used tools to identify SNPs are SAMtools/bcftools and GATK (*Li et al., 2009*; *McKenna et al., 2010*). These tools were developed with the assumption to detect variants within diploid organisms, which limits their detection power for haploid prokaryotes. Therefore we also considered the more generic tool VarScan2 (*Koboldt et al., 2012*), which can predict SNP frequencies in low and high coverage data and some specialized tools for variant prediction within high coverage data, such as LoFreq-Star (*Wilm et al., 2012*), Breseq (*Barrick et al., 2014*) and FreeBayes (*Garrison & Marth, 2012*). Here we used Lofreq-Star, as a previously published evaluation showed it to be superior to Breseq in terms of sensitivity (*Wilm et al., 2012*). We also evaluated FreeBayes which is widely used in Pool-seq experiments for eukaryotes with known pool size but can also analyze a bacterial population with unknown pool size. The tools all work on read alignments or mpileup files and use read and mapping quality scores as well as strand bias filters to reliably detect SNPs. In addition SAMtools/bcftools and VarScan2 and FreeBayes can also be used to identify small InDels. Pindel (*Ye et al., 2009*) uses a pattern growth algorithm to detect small and large InDels from 1 bp up to 10 kb. Large InDels and structural variations (SV), such as translocations, duplications and inversions, are detected by Breakdancer and Delly (*Chen et al., 2009a*; *Rausch et al., 2012*), as they make use of insert size deviations, paired end information and split read information to find variations larger than 300 bp. As an alternative, Cortex_var (*Iqbal et al., 2012*) does not rely on mapped reads but uses *de novo* assembled contigs, which are compared to each other or to a reference in order to identify most kinds of variants. All those approaches have been designed for different degrees of zygosity, ranging from diploid genomes to multiploid populations with low abundant genotypes.

The genotyping of prokaryote populations in experimental evolution experiments is typically based on many NGS datasets with high coverage. There is therefore a demand for fully automated software for read mapping and variant calling, which is both sensitive and accurate, aware of low abundant subpopulations, and which considers all possible types of variants. To the best of our knowledge, no such software workflow has been published so far. In this study we have evaluated variant callers on synthetic data in order to determine and compare their sensitivity and accuracy. This allowed us to develop and validate VarCap, a workflow for accurate and sensitive genotyping of prokaryotic populations. Finally, we applied VarCap to a long-term experimental evolution experiment of a bacterial symbiont of amoebae.

## METHODS

### Creating synthetic variant genomes

Ideally, the organism selected for simulation should exhibit generic properties that make the results applicable for most prokaryotes. In our simulation and evaluation of the variant detection prototype, however, we decided to pick the non-model organism *Protochlamydia amoebophila.* It offered the unique opportunity to experimentally validate variant predictions immediately during the software development. In addition, *P. amoebophila* exhibits

typical properties as its genome size of 2.4 Mb is close to one of the main peaks in the bacterial and archaeal genome size distribution (*Koonin & Wolf, 2008*). For validation purposes, we additionally used 6 different organisms that we selected to represent the diversity of prokaryotic genomes regarding G + C contents and genome size.

Variant datasets were created by randomly inserting different types of variants into reference genomes downloaded from the NCBI Refseq database (*Pruitt et al., 2012*) (Table S1). We used a SNP/InDel ratio of 10 for small InDels and 20 for all InDels, as SNP/InDel ratios for bacterial genomes were often reported between 15 and 20 (*Moran, McLaughlin & Sorek, 2009*; *Chen et al., 2009b*). We also included large InDels, because large insertions hereby also mimic the process of horizontal gene transfer. As structural variations are reported to be crucial for bacterial genome evolution, we also added few translocation, duplication and inversion sites to challenge the detection software.

We created mixed types of datasets containing 135 variations, as well as datasets containing one specific type of variant. The 135 variants of the mixed type dataset consisted of 100 SNPs, 10 small InDels, 10 large InDels and five translocations, five duplications (including one double duplication) and five inversions (Set: sim_135VAR, Table S1).

The 100 SNPs were placed as single SNPs and mutation hotspots. Therefore, the SNPs were positioned as single seeds, to which the other SNPs were randomly assigned with decreasing probability. The maximum number of SNPs within a hotspot was four, which were randomly placed within a distance of 4–60 bases. The size of the large InDels was randomly chosen between five and 2,000 nucleotides, while the size of translocations, duplications and inversions varied from 300 to 2,000 nucleotides. The datasets harboring only one type of variant contained either 100 SNPs, 100 small InDels, 100 large InDels, 50 translocations, 50 duplications or 50 inversions (Sets sim_100SNP, sim_100IndS, sim_100IndL, sim_50ITX, sim_50DUP, sim_50INV).

ALFSim is a genome evolution simulator and was used (*Dalquen et al., 2012*) to simulate the evolution of more distantly evolved subpopulations. Therefore, coding and intergenic nucleotide sequences according to the genome annotation were extracted from the genome reference fasta file. This extracted sequences served as input for ALFSim. From the ALFsim output, we selected a simulated subspecies having a nucleotide dissimilarity of 0.8% resulting in 21,000 SNPs, 100 InDels and three duplications. The resulting fasta file was used for read simulation, construction of a heterogeneous population and prediction of variants.

## Sequencing read simulation

We used SimSeq (https://github.com/jstjohn/SimSeq), version from 4.12.2011, (*Earl et al., 2011*) and pIRS (*Hu et al., 2012*) for the simulation of 100 nucleotides (nt) paired end Illumina reads. The reads were simulated with an insert size of 250 nt and an insert size standard deviation of 10, 20 and 30%. For pIRS we used the supplied error model, while for SimSeq the updated empirical error models for forward and reverse strand were used (hiseq_mito_default_bwa_mapping_mq10_1_Corrected.txt, hiseq_mito_default_bwa_mapping_mq10_2_Corrected.txt). We simulated minor allele frequencies (MAF) by mixing

simulated reads from the original reference with simulated reads from the variant datasets. We created MAF of 40, 20, 10 and 4%.

## Sequence read processing and mapping

The quality of the simulated reads was determined using FastQC (v0.10.0, *Patel & Jain, 2012*). The quality filtering and trimming of the simulated and the sequenced reads was done by Prinseq-lite (0.19.5, *Schmieder & Edwards, 2011*) and Trimmomatic (0.32, *Bolger, Lohse & Usadel, 2014*) and applied with the following settings: first a sliding window with size 10 removed any bases with lower quality than 20 starting from the 3′ side by cutting off the read part containing the low-quality bases. The sliding window approach has the advantage that low quality bases are also removed within the read and not only at the end (which is done, if read trimming is done only from the 3′ of 5′ end). We removed all reads shorter than 40 nt. To remove low quality reads, we discarded any read with an average Phred score below 30. Only read pairs were kept. These reads were mapped against the reference genome using bwa-mem (bwa-0.7.5a, *Li, 2013*; *Li & Durbin, 2009*) with standard settings and stored as bam files. For conversions from sam to bam files and from bam to fastq files (as Cortex_var input), we used SAMtools (0.1.18, *Li et al., 2009*) and Picard Tools (v1.92, http://picard.sourceforge.net/).

### Mapping artifacts

In order to emulate mismapped reads due to an incomplete reference genome, we mapped reads that were generated from an updated (newly assembled) reference genome back to the older and about 20 kB shorter version and to the current version. This dataset did not contain any simulated variants.

### Variant calling

In order to assess true and false positive variant detection rates, artificial non-clonal populations containing SNPs, InDels and SV at abundances of 40%, 20%, 10% 5% and 2% were simulated. We used SAMtools/bcftools (0.1.18, *Li et al., 2009*), GATK-lite (Genome AnalysisTKLite-2.2-8, *McKenna et al., 2010*), VarScan2 (2.3.6, *Koboldt et al., 2012*), LoFreq (0.6.1, *Wilm et al., 2012*) and LoFreq2 (lofreq-star 2.0.0 beta 1, https://github.com/CSB5/lofreq). For the detection of small InDels we used VarScan2 and Pindel (024t, *Ye et al., 2009*). For large InDels and structural variations (SV) we used Pindel which is described to work well between on variations between 1 and 1,000 nt, breakdancer (breakdancer-1.1_2011_02_21, *Chen et al., 2009a*) and delly (0.0.11, *Rausch et al., 2012*) (both start calling SV at 300 nt). Additionally, we used the assembler cortex_var (CORTEX_release_v1.0.5.14, *Iqbal et al., 2012*), which can detect variations by comparing assembled contigs to a reference genome sequence. The sensitivity and precision of the combined workflow were calculated as: sensitivity = TP/(TP + FN), and precision = TP/(TP + FP). The TP, FP and FN are measured per variant, giving e.g., a SNP and a large deletion event the same weight.

## Setting the minimum abundance for a variant

In order to call a variant, it has to be present within a minimum count of sequencing reads. Some variant callers need a variant to be present on 4–8 reads, so we set eight reads as the

minimum absolute abundance (MAA). However, as read coverage slightly varies along the genome, we also used minimum relative abundance (MRA), which is the percentage of variant reads compared to the total coverage. So, a MAA of eight reads corresponds to a MRA of 2% at 400× total coverage.

## Examining the similarity of repetitive regions

We used the edit distance in order to measure the similarity of repetitive regions. The edit distance measures the similarity of two sequences by counting the differences between them. This difference can be a substitution, insertion or deletion of a nucleotide. Therefore, an edit distance of one means that two sequences differ in either a substitution, insertion or deletion of a nucleotide.

## Analysis of a long-term experimental evolution experiment

We applied the VarCap workflow to a long-term experimental evolution experiment in order to evaluate its performance on Illumina PE data. Two independent laboratory cultures of the amoeba symbiont *Protochlamydia amoebophila* were subjected to NGS sequencing using the Illumina Genome Analyzer II platform (100 bp PE reads, 250 bp insert size, 3,000× coverage, 250 bp insert size) about nine years after its genome was initially sequenced by Sanger sequencing (*Horn et al., 2004*) (SRA: SRR5123091). For analysis, the obtained Illumina reads were randomly split into replicate read packages with 250-fold coverage each and utilized to detect variant sub-populations at different abundances.

## PCR verification of variations

To verify the variations at positions 1339224, 1339720, and 1338568 in the genome of *P. amoebophila* we amplified the region 1338371-1339843 by PCR using the primers LS0003 5′-AGCTGCATCATTTATCTTCTAG-3′ and LS0004 5′-ATCAGTCCACCTACTATCATG-3′. The obtained 1,472 bp fragment was cloned into the pCR4-TOPO vector (Invitrogen). Subsequently, 16 of the obtained colonies were picked, and the presence of variations in the cloned amplicons was checked. Clones were sequenced by Sanger sequencing with the primers T3 and T7. Similarly, 14 putative variations in a repetitive region between positions 1533689 and 1534636 were assessed using the primer pair LS0005 5′-TCTCTAGCTCT TTCGCAAATTG-3′ and LS0006 5′-CAGTGTTTAACTGGCTGAAAC-3′.

## A Galaxy instance of VarCap

We simplified the use of VarCap for non-experts to a 3-step process facilitated by our Galaxy server (*Afgan et al., 2016*): (I) Create account and login, (II) Upload your data to Galaxy and (III) Run the VarCap workflow. After the workflow is finished, the user is informed via Email notification. The results are viewable at and downloadable from the website. The output files consist of a VCF file with a detailed description of the variants as wells as two PDF files, which contain overview information about variant and total coverage and frequency information.

# RESULTS

## Determination of methods capable of sensitive detection of low abundant variations

### Evaluation strategy

At the moment, there is no software tool or method that could detect all different types of variants simultaneously which are relevant for prokaryotic genomes. Therefore, we separately evaluated variant detection tools for SNPs, InDels and structural variants (SV). Representative methods for these three targets were selected according to their underlying methodologies. In order to identify the variant calling tools that most sensitively and reliably detect low abundant variant, we initially utilized our most basic variation model (sim_135VAR). It incorporates examples of the typical and expected types of variations in microbial genomes, located in typical distances to each other. From these results, we constructed a preliminary software framework, which was used as basis for the further evaluations and improvements.

### SNPs

Among the many available SNP calling software tools we have compared LoFreq-Star, Varscan2, GATK, SAMtools/bcftools, FreeBayes and Cortex_var. All of these tools, except Cortex_var, rely on the mapping of reads to a known reference. Cortex_var, instead, *de novo* assembles variant reads into contigs and thereby detects SNPs. SAMtools/bcftools and GATK were only designed for homozygous and heterozygous genomes (*Yost et al., 2013*), whereas LoFreq-Star, Varscan2 and Cortex_var should be able to detect low frequency variants from high coverage sequencing data. Variants were simulated at minor allele frequencies (MAF) of 40%, 20%, 10% and 4% and evaluated at minimum relative abundance (MRA) cutoffs of 20%, 10%, 5% and 2% accordingly. This means that ideally all variants present at and above those frequencies should be detected. At MRAs of 20% and 10%, variants were detected by all SNP calling software tools at a similar sensitivity (Fig. 2A). According to the expectations, the detection rate of GATK and SAMtools/bcftools was worse compared to the other programs when the MRA was reduced to 5%, 2% and 1% (Fig. 2A). At a low MRA of 1% LoFreq-Star shows less sensitivity than Varscan2. This is to be expected, as LoFreq-Star generates its own detection threshold based of coverage and quality to avoid FP and therefore detects no variants below that threshold (Fig. 2A). The price of the higher sensitivity of Varscan2 at MRA of 1%, however, comes at the price of elevated FP variant predictions. FreeBayes was able to detect variants at all MRAs with similar sensitivity (Fig. 2A). However, we observed FP at MRAs of 2% and 1% and therefore did not include this tool in further analysis.

### InDels

Varscan2 and Pindel were used for the detection of small InDels, and Pindel, Breakdancer, Delly and Cortex_var for the detection of larger InDels. For small InDels, the MSA approach used by Varscan2 should perform at a similar rate as the pattern growth algorithm used by Pindel. Pindel, however, is designed to detect InDels from 1 to 10,000 bp as it uses a mapping/pattern growth/split read approach. Therefore, it should be able to detect the

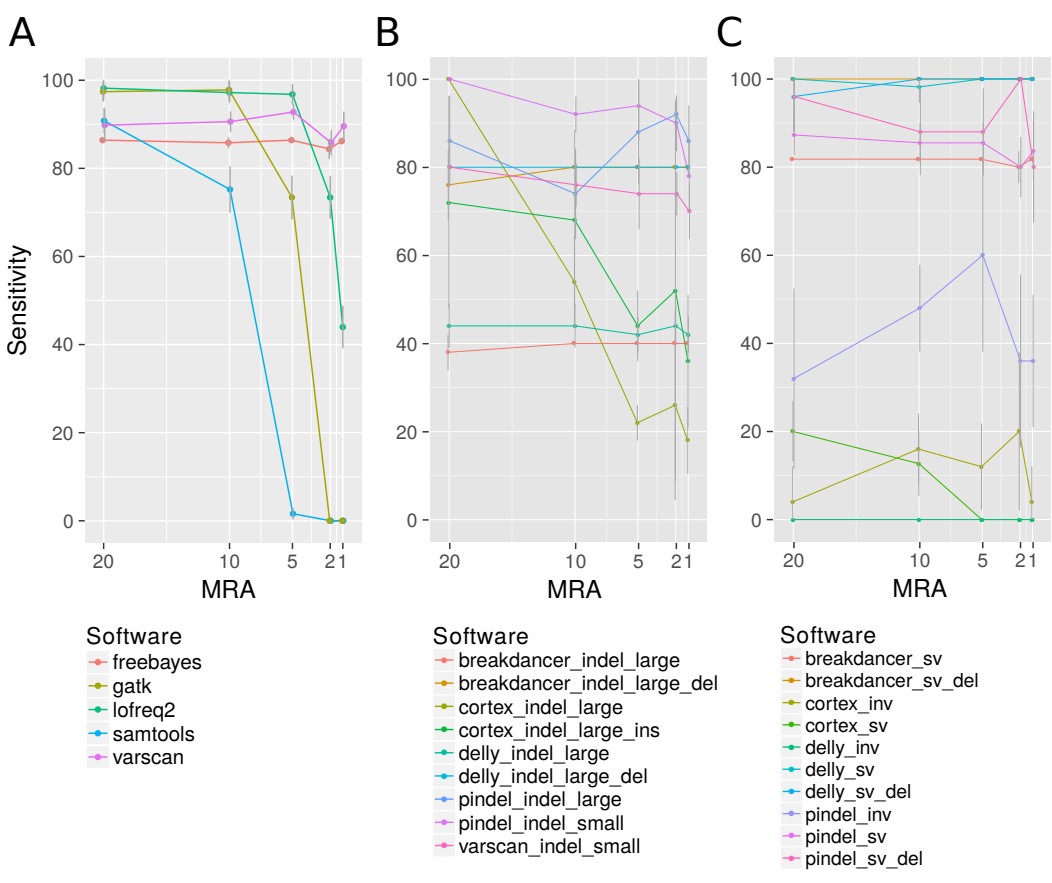

**Figure 2** **Analysis of the detection rate of variants with regard to Minimum Relative Abundance (MRA), variant type and different variant calling software.** It shows the detection rate of different SNP (A), InDels (small denotes small InDel, B) and SV callers (C) with respect to the MRA frequencies of 20, 10, 5, 2 and 1%. For Breakdancer, Pindel, Delly, and Cortex, two values are given: detection rate of all InDels and specific detection rate for deletion or insertion only.

positions of small and large InDels with base pair precision. Breakdancer and Delly are designed for the detection of InDels larger than 300 bp. They use paired end read information for InDel detection, therefore the position of the large InDels may not be reported at bp resolution. Cortex_var is expected to be less sensitive because of the *de-novo* assembly approach, however it can supply more information than the mapping approaches, including e.g., position, length and sequence of an insertion.

The detection rate of InDels showed little effect to different MRA values (Fig. 2B) (except SAMtools/bcftools, see discussion above). Instead, the sensitivity is related to the methodology underlying the software. We observed that Varscan2 can only detect very short InDels (1 bp) with the same sensitivity as Pindel, which detected all sizes of InDels with high precision. According to our expectations Breakdancer should have a diminished detection rate for large insertions, as it only considers information about insert size deviation of paired reads and regions with an increased number of anomalous read pairs. We found, that it detects 100% of all large deletions but misses all insertions. As expected, the assembly method used by Cortex_var performs inferior compared to the others. However, it was

one of the only two tools that were able to detect large insertions. It detected one third of the large insertions and reported the inserted sequence, while Pindel detected the position of large insertions at a higher rate, but without revealing any details.

## Structural variations (SV)

For the detection of SV, we used Pindel, Breakdancer and Delly, and we added Cortex_var specifically for inversion detection These programs differ slightly in their methodological approaches. We expected Delly to be superior to Breakdancer because of the additional split read alignment. Moreover, we expect a limitation of Pindel at larger rearrangements, because the pattern growth algorithm is used within defined limits (up to 10 kb). All tools should be able to detect inversions; however, they are reported as being harder to detect than other SVs. Breakdancer and Delly detected SV, like duplications and transpositions, regardless of the MRA with high sensitivity (>90%). As expected, the detection rate of Pindel is lagging behind (80%) according to of the suggested internal limits of 10 kb. However, the pattern growth method of Pindel was more precise in terms of position and length of the SV as it always hit the exact starting position while Breakdancer and Delly can be off up to 70 bases (Fig. 2C). We additionally found that large InDels were called at the sites of translocations events (Fig. 2C). This is not entirely unexpected, as a translocation consists of an excision and the consecutive insertion of the excised genomic fragment. The excision can also be seen as a deletion of a fragment and is therefore a partial detection of a more complex type of variant.

Inversions, however, could only be detected at a minor fraction as break positions by Pindel (70% as break positions) and as inversion by Cortex_var (10%) (Fig. 2C inv).

## Selected software tools for VarCap

We use LoFreq-Star and Varscan2 for SNPs and Varscan2 and Pindel for small InDels for composing VarCap because they showed similar sensitivity although using different methodological approaches. For larger variants or SV, we observed that a combination of pattern growth, split read and paired end read information approaches, which are used by Pindel, results in high sensitivity. This method works well within defined limits (1–10 kb). By using only paired end information (Breakdancer), it is possible to detect larger variants at the cost of a lower length limit (300 bp) and a coarser resolution of the variant position. Cortex_var, however, was inferior in sensitivity but revealed more information about the detected variants by using a *de-novo* approach. This information can be used to correctly identify the type, position, length or sequence of the variant. Therefore, we use Pindel, Breakdancer and Cortex_var for large InDels and Breakdancer, Delly, Pindel and Cortex_var for SV.

Due to the different variant calling abilities of the different tools at low frequencies, we combined different tools to increase the sensitivity (Fig. 3A). Beyond sensitivity, we also monitored the precision of the different tools for each type of variant in order to avoid methods that have excessive numbers of FP (Fig. S1). As a consequence, Cortex_var was used to predict InDels and inversions but not for SNPs as it accumulated many false positive SNPs in certain areas at low frequencies. We also discontinued to use FreeBayes

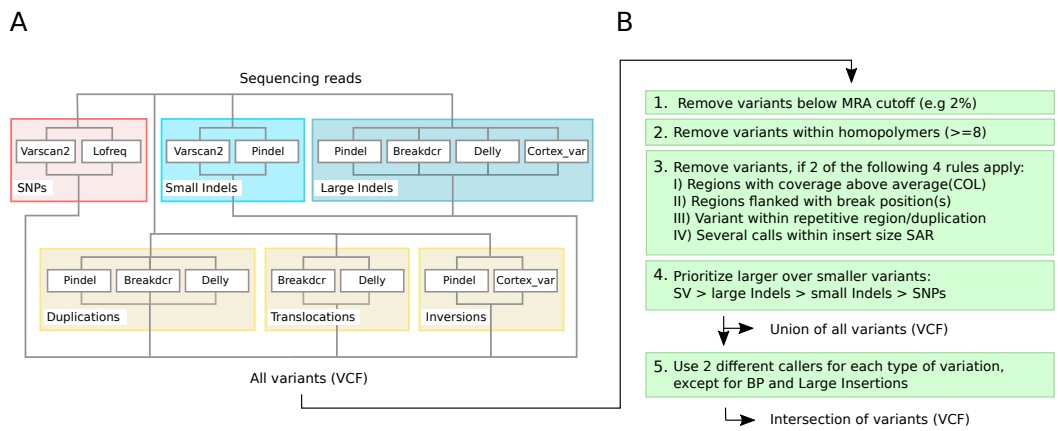

**Figure 3  Detection capabilities of different tools at low frequencies.** (A) shows the variant types that were successfully detected by the different software tools while (B) shows the post filtering steps to eliminate false positives. The post filtering step generates 2 output files: one file includes the union of all variants, while the other contains the intersection of variants (except break positions (BP) and large insertions, which are also reported as single calls).

for SNP calling, as it showed low precision at MRAs of 2% and 1%. Taking together all selected software tools, we were able to detect all variants, except inversions, at a MRA of down to 2% with high sensitivity (Fig. 4).

## VarCap—a variant calling workflow with high sensitivity and specificity

### False positives due to sequencing errors

False positives occur due to sequencing errors, which are typically present at and below a rate of 1%, therefore we expect them to cause FP calls at and below this relative abundance. In order to study the influence of sequencing errors on different software detection tools, we analyzed seven differentially composed samples and focus on MRAs of 2% and 1% as this seems to be the critical boundary for FP prediction (mono_02-07). At a MRA of 2% we observed a false positive rate for SNPs, small InDels and Duplications of 0.5 to 1 FP per Megabase (Mb) (Fig. S2B: MRA 2). At a lower MRA of 1%, we observed an increase in FP (Table 1). At an MRA of 1%, we could nearly completely find all types of variants, except inversions, which we could identify at a rate of 95%. However, the false positive rate for SNPs increased to 80 FP per Mb, while the FP rate for other variants stayed below one FP per Mb (Figs. S2A, S2B: MRA 1). This clearly demonstrates that false positive SNPs are caused by sequencing errors, while the other types of variants stayed at the low rate (∼1FP/Mb).

In order to get more insights about the other FP, we examined them in detail at both MRAs. We found that FP of small InDels locate within repetitive regions of the genome. These regions are almost identical areas of the genome at a size that is longer than the insert size of the reads and have an edit distance of three or less. Due to their similarity, variant reads can be mapped to similar regions and cause FP calls there.

In order to evaluate how MAA and coverage influence the FP rate, we simulated sequencing coverage from 80 to 1600× (using the sim_135VAR dataset) and used MAAs from four to 20 to remove FP from the unfiltered variant predictions (Fig. 5). For each

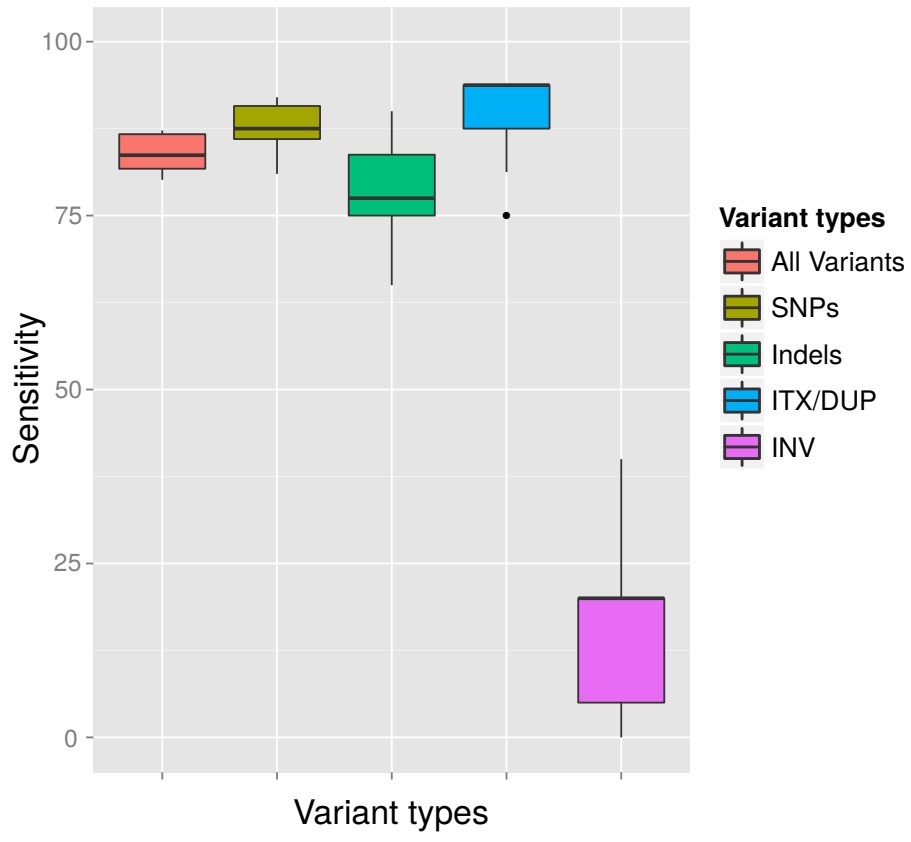

**Figure 4** **Average detection rates and standard deviation of all callers for variants simulated at a MAF of 4% and a MRA cutoff of 2%.** The variant detection rates are shown in percent for all variants (ALL), only SNPs (SNP), only InDels (IND), duplications and translocations (ITX/DUP) and inversions (INV). The results show the expected sensitivity of VarCap, as we use a MRA of 2% as a default setting to avoid false positives.

**Table 1** **Detection sensitivity and precision of the combined workflow for a different number of callers and at different simulated minor allele frequencies (MAF) and minimum relative abundance (MRA) cutoffs.** The table shows the numbers for the observed true positives (TP), false negatives (FN), false positives (FP), sensitivity and precision of the combined workflow at MRAs of 10, 5, 2 and 1% under the requirement that either one or two callers (Min Caller) had to confirm each variant.

| | | | | | | | |
|---|---|---|---|---|---|---|---|
| 1 | 20 | 10 | 139 | 2 | 0 | 0,986 | 1 |
| | 10 | 5 | 137 | 4 | 0 | 0,972 | 1 |
| | 4 | 2 | 138 | 3 | 0 | 0,979 | 1 |
| | 4 | 1 | 141 | 0 | 1,238 | 1,000 | 0,102 |
| 2 | 20 | 10 | 135 | 6 | 0 | 0,957 | 1 |
| | 10 | 5 | 133 | 8 | 0 | 0,943 | 1 |
| | 4 | 2 | 133 | 8 | 0 | 0,943 | 1 |
| | 4 | 1 | 135 | 6 | 0 | 0,957 | 1 |

**Figure 5  Influence of total coverage and MAA on FP rate.** The table numbers show the FP per Megabase in context to coverage and MAA while the different colors indicate the corresponding MRA levels. We simulated MAFs of 20, 10, 4, 2 and 1% (using the sim_135VAR dataset) and detected at MAA cutoffs from four to 20 bases to support a variant.

coverage/MAA setting. we report the resulting calculated MRA. We report the FP rate as FP per Mb, as with this normalization step we are independent of the genome size. Otherwise a 4 Mb sized genome would produce twice as many FPs as a 2 Mb genome. We detected, that it is necessary to use an MAA cutoff in addition to an MRA cutoff to avoid FP calls at lower coverages (Fig. 5, see FP counts at MRA2 at coverage 160×).

## FP due to mismapped reads

Mismapped reads have been reported as the cause of FP (Li, 2014). Therefore, incomplete reference genomes lead to reads getting mapped to similar regions and cause FP calls there. To review this finding at a MRA of 2%, we mapped reads without variants back onto an artificially shortened reference genome. We observed ∼180 FP SNPs/75 FP per Mb which were present at different abundances (20%, 8%, 3%) and grouped into hotspots (Fig. 6A). False positive variants were not observed when mapping the reads to the correct reference (Fig. 6B). This finding strongly supports our assumption that wrongly mapped reads cause FP variant calls. A closer investigation of the relevant regions revealed the presence of neighboring break positions, which may indicate both: either a larger structural variation or mismapped reads due to an incomplete reference genome.

To identify possible false positives due to mismapped reads, we implemented the following filtering steps: As suggested in prior discussion of this topic (Li, 2014) we used the coverage information at the variant sites to tag possible false positives. However, coverage information alone is too coarse for the resolution of low frequent FP. Therefore, we additionally monitor break positions that flank or reside at the variant positions to identify regions with mismapped reads. As all FP were present as small clusters or hotspots, we tagged regions that hosted more than 4 SNPs within a sliding window at the double length of the insert size and were accompanied by a break position (BP) as possible FP causing regions. With the application of these filters we could identify and exclude the FP calls (Fig. 6C).

A closer look at inversions revealed that they were mostly not identified as inversions, but the start and the end point of the inversion were marked as break positions (Table S2). Break positions occur because only one read of a pair can be mapped, leading to an accumulation of only forward or reverse reads. They indicate a larger sequence difference between the reads and the reference and are therefore a more general indicator of a larger structural variation. Therefore, these calls represent a partial resolution of the variant.

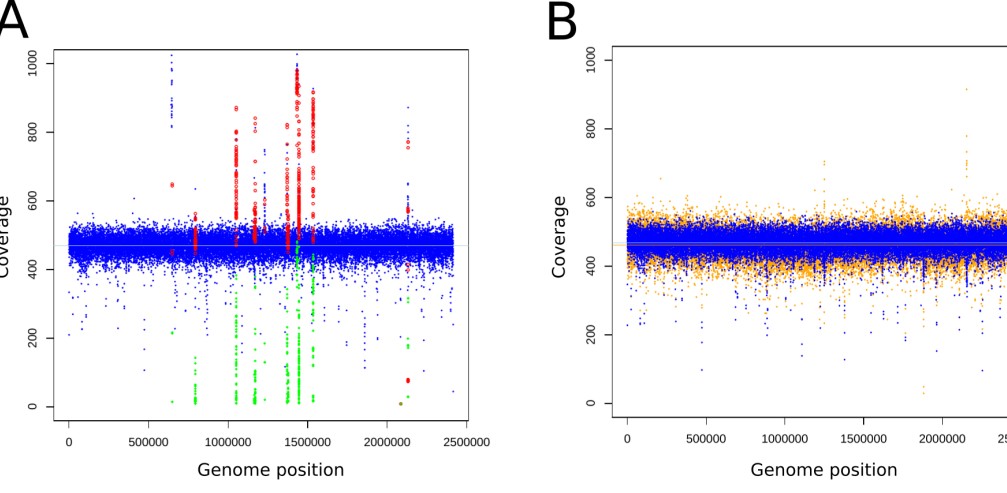

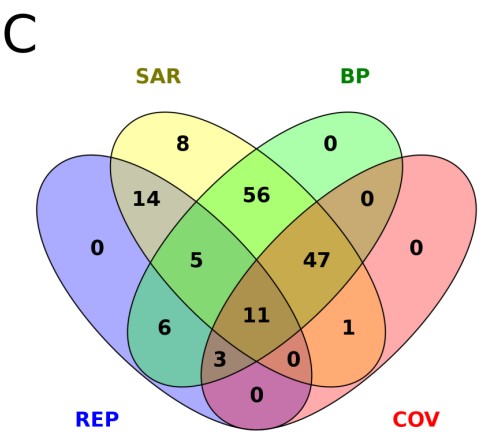

**Figure 6** **Coverage plots of simulated and re-sequenced data.** The simulated reads without variants were mapped back to an incomplete reference (A) and the complete reference (B). The blue circles denote the total coverage along the genome while the green diamonds show the coverage of the FP variants and the red circles the total coverage at the FP positions. As a comparison, we show the coverage distribution of sequenced reads against the complete reference in orange in the background of (B). The coverage peaks at 1,220,000 and 2,150,000 are due to additionally mapped mitochondrial reads. The light blue and orange lines show the average coverage distribution along the genome. A total of 149 of 154 of the FP from 6A could be tagged and filtered by the properties coverage (COV), within repetitive region (REP), within SNP accumulating region (SAR) and located close to a break position (BP) as shown in (C), the remaining five were single calls and thus eliminated by the constraint of two callers per variant.

In order to identify and exclude false positives we apply the following filters: to avoid FP SNP calls caused by sequencing errors we apply a MRA of 2%. To avoid FP due to reads mapped to repetitive regions, we mask nearly identical regions according to the properties described above within the reference genome and tag variants that are found within these regions. In order to resolve FP that are caused by incomplete detection of the true variant type, we prioritize larger over smaller variants. Therefore, we assign smaller variants to larger ones, if they describe a component of the whole variation; for example, large InDel at excision site of translocation.

## Performance of combined post-processing and filtering in VarCap

We observed that a gain in variant calling sensitivity decreased the precision. Therefore, we added a post-filtering step to the workflow in order to eliminate possible FP. We incorporated a post-processing step for each variant that aims to eliminate FP due to sequencing errors, repetitive regions, partially detected variants and mismapped reads due to reference incompleteness. As a consequence of the dissimilar variant detection rates of some methods, we decided to use more than one tool for each type of variant. In order to gain precision and robustness, for high confidence variants, we required an intersection of predictions per variant. Therefore, a variant call had to be supported by at least two different tools. This step further contributed to an improved precision at low MRA cutoffs (1%), while the detection rate was only slightly diminished (Table 1). This finding is backed up by a recent publication, which made a similar observation regarding the intersection of different tools (*Kofler et al., 2016*).

## Genotyping of diverse synthetic prokaryotic populations
### Detection rates in different genomes

Genomes exhibit different properties, such as G + C content and size, which could potentially affect the sensitivity and accuracy of variant calling. Therefore, we evaluated our variant calling workflow on six different genomes. These organisms consisted of five bacteria and one archaeon, with differing G + C content ranging from 26 to 72% as well as a differing genome size ranging from 0.68 to 8.66 Mb. The workflow was used with a MRA of 2% as well as at a MAA of eight reads supporting a variation. In concordance to our previous results we could detect most of the (simulated) variants (>90%). However, at a MRA of 2% we could not observe any dependency on G + C content or genome size while the MAA of eight reads resulted in fewer variant detections at high G + C content and genome size (Fig. 7). This observation confirmed our previous observations to use a MRA as a general minimum cutoff for variant detection as it showed little influence to different genome properties. This, however, does not remove the need for a fixed MAA in case of low coverage regions.

## Detection rates in a distantly evolved population

More distantly evolved populations may lead to a higher number of variants if they are under positive selection. This could affect the sensitivity of variant calling. Therefore, ALFSim (*Dalquen et al., 2012*) was used to simulate a more distantly evolved population by integrating evolutionary changes (SNPs, InDels and duplications) into the *P. amoebophila* genome. The evolved genome showed a similarity to the reference around 99%, as it contained around 21,000 SNPs, 100 InDels and three gene duplications.

We evaluated the sensitivity of the variant calling by VarCap at a low abundant subpopulation of 4%. We used a MRA of 3%, 2% and 1% as well as a MAA of eight reads (equals a MRA of 2% in a 400× covered genome). Depending on the minimum abundance requirements, we were able to detect between 90% and 99% of all SNPs, between 74% and 94% of all InDels and two out of three duplications. The true positive detection rate of SNPs increased to 98%, while the false positive rate remained below 0.3% when lowering the MRA from 3 to 2%. However, if we lowered MRA further to 1%, we increased the TP

A

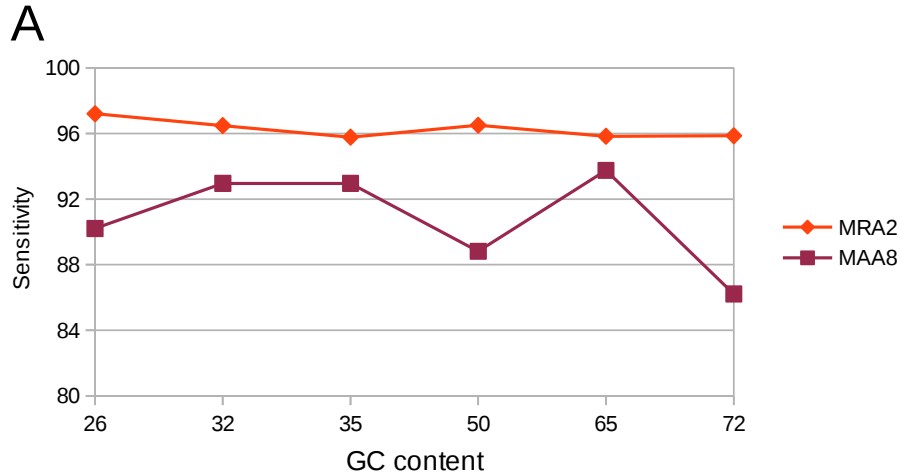

B

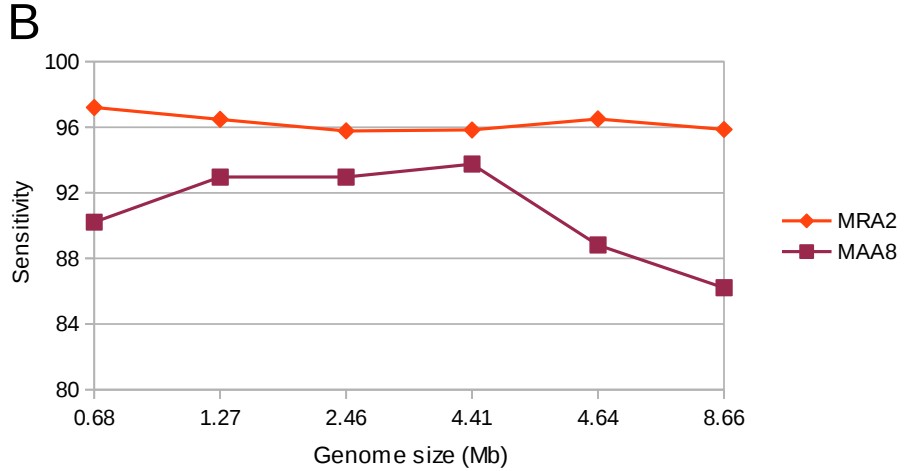

**Figure 7** **Detection rate of variants in various genomes at minimum absolute and relative abundance.** The observed percentage of True Positives is shown for six organisms with differing GC content (A) and genome size (B). The total coverage is at 400×, the coverage of the subpopulation containing 135 variants is at 16×. No False Positives were observed at the MAA of eight and MRA of 2%.

rate to 99% while augmenting the FP rate close to 400 FP/Mb (Fig. 8A). At a MRA of 2% we could locate most FP within repetitive regions and recent duplications (Fig. 8B), while at a MRA of 1% we detected mainly FP caused by the sequencing error rate (Fig. 8B). At a MRA of 2%, we were able to detect over 90% of all InDels including all small InDels (size = 1), without experiencing false positives (Fig. 8A). With regard to duplications we were able to find two of them at most MRAs, while missing out the shortest one constantly (Fig. 8C SV(DUP)). These findings confirm that we are able to achieve a high accuracy even if the evolved genomes are rather dissimilar. However, a novel finding was that also recent duplications can lead to wrongly placed reads as they are similar to repetitive regions. Therefore, we also included tagging of duplicated regions as possible regions for FP calls into our workflow.

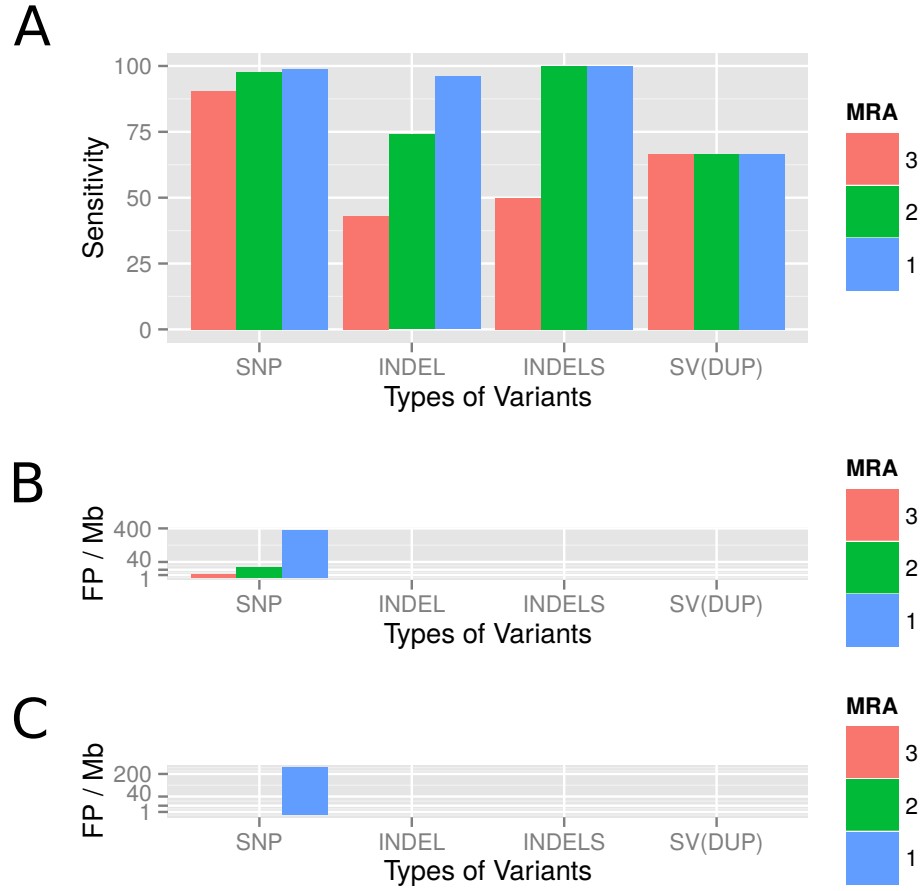

**Figure 8** **Observed detection rates of variants which were simulated using a genome evolution soft-ware (ALFSim) and detected at different minimum abundances.** Simulated variants included SNPs, small InDels (INDELS), large InDels (INDEL) and structural variations (SV) as duplications (DUP). (A) shows the sensitivity at MRAs of 3%, 2% and 1%. (B) shows the False Positives for SNPs as counts per Megabase at the different MRAs. At these minimum abundances, no FP for InDels and SV were detected. (C) shows the FP per Megabase after filters have been applied. SNP, Single nucleotide polymorphism; IN-DEL, Large InDels; INDELS, Small Indels (<10nt); SV(DUP), Duplication.

## Detecting variants in a real bacterial population after long term cultivation

In order to predict variant frequencies within an evolving population, the variant calling workflow was applied to a long-term cultivation experiment of *P. amoebophila*. Different MRA cutoffs from 20% to 2% were used and revealed that variants were present at frequencies down to 2% (Fig. 9A, outer rings). Variants within repetitive regions (Fig. 9A, inner connective lines) were tagged for further inspection. At a MRA of 2% we observed a total number of 71 variants, which comprised of 34 SNPs, 20 InDels and 17 structural variants. The SNPs and small InDels were annotated using SNPEff (*Cingolani et al., 2012*). This revealed, that around 83% of them were situated within coding regions (Table S3). At a MRA of 2% we could find three InDels present at a MAF of 2% and one InDel at a MAF of 3%, which were located within homopolymeric regions of length 10 or longer (Table S4). Thus, those InDels were tagged as probable FP for further manual inspection.

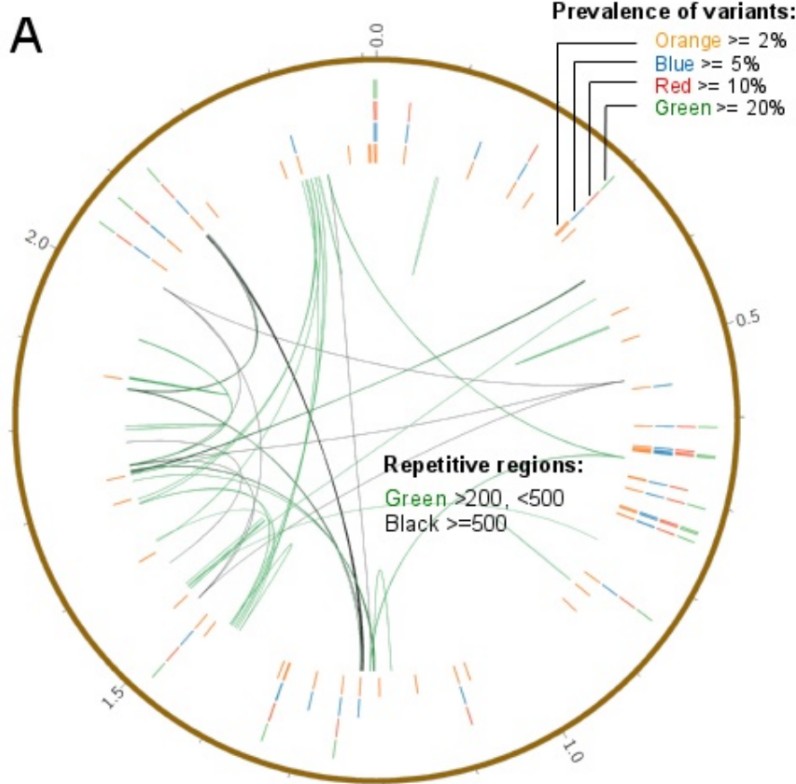

Genome map of *P. amoebophila*

| | MRA 2 | MRA 5 | MRA 10 | MRA 20 |
|---|---|---|---|---|
| SNP | 34 | 23 | 16 | 11 |
| Indel | 20 | 8 | 8 | 5 |
| SV | 17 | 9 | 8 | 8 |

Variant counts

**Figure 9  Prevalence of variants within a long-term culture with respect to their MRAs.** (A) shows the prevalence of variations at MRAs of 20%, 10%, 5% or 2%, which are visible in the four differently colored outer circles and the presence of repetitive regions within the reference genome (inner connective lines). (B) shows a more detailed view of the number of variations found at MRAs of 20%, 10%, 5% and 2%.

For the validation of the variant calling prototype of VarCap we picked three variations for further analysis that were present at abundances of 4%,11% and 28%, accordingly. We performed PCR of the regions surrounding the three variants, cloned the fragments into vectors and picked 16 clones of each variant for Sanger sequencing (Table 2, Files S1–S2). We were able to detect all three variants and thus could confirm the predictions of the VarCap software.

**Table 2 Experimental validation of a subset of the predicted variations.** Three variant positions at different frequencies were amplified by PCR, cloned and Sanger sequenced for validation.

| Position | Frequency | Clones total | Clones supportive | Sanger confirmed |
|----------|-----------|--------------|-------------------|------------------|
| 1,338,568 | 28 | 16 | 6 | Yes |
| 1,339,720 | 11 | 16 | 2 | Yes |
| 1,339,224 | 4 | 16 | 1 | Yes |

## DISCUSSION

Population genomics of microbes is most powerful if we meet the challenge of detecting all types of genomic variations even at low frequency. We therefore developed, evaluated and validated VarCap, a workflow that allowed us to reliably identify variants even within low abundant alleles.

### Increasing sensitivity

We tested the capabilities of the relevant variant calling tools and observed substantial sensitivity differences between the different methods. In order to improve the overall sensitivity, we decided to integrate different tools for variant detection into a combined workflow, in which every variant can be detected by more than one caller.

### Increasing precision

As more tools are likely to introduce more errors, we also optimized the overall precision. Detecting sequencing errors and mismapped reads was key to control the rate of false positives. When dealing with sequencing errors, we observed that for SNP detection a MRA cutoff of 2% was sufficient to keep a safety margin to false positives appearing at a MRA cutoff of 1%. Although we found, that a relative abundance cutoff (MRA) is superior to an absolute cutoff (MAA). We also observed that a MRA cutoff leads to FP if the read coverage is too low ($<200\times$). Therefore, we also apply a fixed MAA cutoff of 8 reads to remove FP at low coverage positions. This implies, that for detecting a subpopulation present at a MAF of $>2\%$ we need a minimum sequencing coverage of $400\times$. Sequencing experiments should therefore aim for at least $500\times$ to account for reads removed by quality filtering and fluctuations in coverage along the genome.

We could not detect any FP InDels within our simulated data but detected several spurious InDels in homopolymer regions of the re-sequencing experiment. These are probably sequencing/PCR artifacts that are not introduced by read simulators. Based on our findings InDels below a MRA of 10% should be tagged as potentially false positive if they are located within a homopolymeric region ($>8$ bases).

Mismapped reads can occur within repetitive regions, undetected duplications, or incomplete reference genomes. Therefore, we flag repetitive regions greater than the insert size in order to mark variants appearing within these regions for further inspection. Unnoticed duplications or incomplete references cause reads to get mapped to similar regions, which can be observed by higher coverage and/or variant accumulation within these areas. In order to overcome false positives by misplaced reads, we removed variants that at least fulfill two of the four following rules: (I) Either variants lie within regions with a coverage

of 20% above the average and/or (II) if there is a break position detected at or within read length of the variant site and/or (III) if they lie within a repeat region and/or (IV) if more than five variants lie within the length of one insert size. The efficiency for FP removal for each rule may differ among experiments as they dependent on organism, experiment setup, sequencing and reference quality. Therefore, we strongly suggest to use all rules in combination for a most flexible removal of FP predictions due to misplaced reads.

### Intersection of predictions

We remove FP caused by sequencing errors, Fp due to homopolymer errors and FP due to misplaced reads for all variant calls generated by the different tools. Furthermore, for extracting high confidence variants, we performed an intersection of different tools per predicted variant. Therefore we requested each variant to be confirmed by at least two callers, except for break positions, inversions and large insertions. Inversions and large insertions are harder to detect than other variants. Therefore, an intersection would further decrease their count. Break positions, on the other hand, do not lead to FP predictions. They just indicate problems in mapping, which can be due to structural variants or incomplete/distant references.

### Limits of variant detection

We observed that insertions and especially inversions were harder to detect than the rest of the variations. This is not unexpected, as current methods for their prediction need sufficient support by reads, which may get lost at low frequencies. In the simulated evolution data, we missed the shortest duplication constantly. This may be related to a combination of callers working at their operational limits (300 bp) and a diverging evolution of the duplicated sequence due to newly introduced SNPs.

According to our results, we could establish rules for filtering out errors and help with the interpretation of different types of variations (e.g., SNP, duplications). Using these rules, we have built a fully automated workflow that reliably predicts rare variants in deep sequencing data.

## CONCLUSION

We created VarCap, a fully automated workflow that allows scientists to rapidly predict variants within high coverage, short read paired end sequencing data. VarCap automatically performs quality filtering, mapping, variant calling and post-filtering of the predicted variants. VarCap can be used for single organism as well as multi organism experiments as long as fASTA references are provided for the involved organisms (in multifasta format). In order to allow a broad community to use VarCap, we implemented VarCap within our Galaxy server, which is publicly available at http://galaxy.csb.univie.ac.at. VarCap includes default parameter settings, derived from our evaluation experiments, to keep it as simple as possible for the user. The estimated runtimes for $2 \times 0.5/1/2$ Gb sized samples are around 35/70/150 min (Fig. S3) on an 8 core/32 Gb RAM virtual machine. The output of VarCap is a VCF file with a detailed description of the variants and two PDF files, which give a graphical overview of variant coverage and their frequency distribution. VarCap is designed

to predict different allele frequencies in experimental evolution experiments, and it is able to detect and report the frequencies of multiple genotypes within clinical samples e.g., multiple infections.

### Funding

This work was funded by a grant from the European Research Council (ERC StG EVOCHLMY, grant no. 281633). The funders had no role in study design, data collection and analysis, decision to publish, or preparation of the manuscript.

### Grant Disclosures

The following grant information was disclosed by the authors:
European Research Council: 281633.

### Competing Interests

Thomas Rattei is an Academic Editor for PeerJ.

### Author Contributions

- Markus Zojer and Lisa N. Schuster conceived and designed the experiments, performed the experiments, analyzed the data, contributed reagents/materials/analysis tools, wrote the paper, prepared figures and/or tables, reviewed drafts of the paper.
- Frederik Schulz and Alexander Pfundner conceived and designed the experiments, performed the experiments, analyzed the data, contributed reagents/materials/analysis tools, wrote the paper, reviewed drafts of the paper.
- Matthias Horn and Thomas Rattei conceived and designed the experiments, analyzed the data, wrote the paper, reviewed drafts of the paper.

### Data Availability

  Github: https://github.com/ma2o/VarCap.
  Galaxy: http://galaxy.csb.univie.ac.at.

### Supplemental Information

Supplemental information for this article can be found online at http://dx.doi.org/10.7717/peerj.2997#supplemental-information.

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
