# Peer review of "Variant profiling of evolving prokaryotic populations"

_PeerJ, doi:10.7717/peerj.2997_

## Round 0.1 · original submission · Major Revisions

While I do not necessarily expect you to add results for FreeBayes and breseq (Reviewer #2), you should at least address the suitability or limitations of these tools.

You may take Reviewer #1's suggestion regarding moving figures to Supplemental Information on advisement, as I do not share the concern.

Please proofread for English grammar to improve readability in places, although the MS overall is clear. There are issues both of grammar (e.g. P5 L17 "was"->"were") and spelling (e.g. "forth" for "fourth" in the legend to Supp Table 2).

Please review the consistency of how terminology is used in the text and figure captions (for example, the capitalization of "Indel" and "InDel" varies throughout).

Please provide definitions in the Figure captions and Table legends of the abbreviations used within them (e.g. all the abbreviations in Fig 5, and in Fig 9 does the "S" in "INDELS" presumably indicates "small"?)

P 5 L2. Please clarify that these 2 files are to be found within the aforementioned github repo.

P6 L31. Please clarify in the MS whether "sensitivity" and "precision" are bring measured at single nt resolution (for variants larger than a single nt).

P12 L19. This appears to be referring to Supp Table 2 rather than Supp Table 1.

P12 line 27. Please clarify what is meant by "accuracy" - do you mean "precision"?

Optional: The organization of the short Discussion is somewhat hard to follow, it could be improved by breaking it into coherent paragraphs.

Optional: I would have expected, in the Discussion, some more consideration of the limitations and possible extensions of both the pipeline and the recommended rules for calling variants.

Table 1. I am concerned that sensitivity is not monotonically increasing with MRA. What am I missing?

Fig 3. Is there an explanation for the intermediate peak at MRA 5 in the cortex results?

Fig 10B. Are these numbers counts? Please clarify in the caption.

Is the last figure in the MS (labeled "Introductory Figures") intended to be included? There is no caption and it is not clear where it belongs in the text.

Supp Table 3 appears to be cut off at the right in the review PDF.

Supp Table 4 would ideally be made available in text format (as it will not be useful as PDF).

Reviewer 1 ·

Basic reporting

Neither simulated nor real sequencing data were made available. I'm not sure if this is required for computational methods/workflows, but at least the real data set (Protochlamydia amoebophila) would be of value to others and should be uploaded to e.g. SRA

Experimental design

No comments

Validity of the findings

I would have wished the authors would have described and motivated
steps and choices more clearly. Many descriptions could be more
explicit, for example, it's not immediately clear whether VarCap
intersects or uses the union of predictions per variant type. An
example is "we combined different tools to increase the
sensitivity" and "Fig 2B: "use 2 different callers for each type of
variation. Inspect single calls"

See also comment on missing data above

Additional comments

I would have wished the authors would have described and motivated
steps and choices more clearly. The descriptions could be more
explicit, for example, it's not immediately clear whether VarCap
intersects or uses the union of predictions per variant type. An
example is "we combined different tools to increase the
sensitivity" and "Fig 2B: "use 3 different callers for each type of
variation. Inspect single calls" So it is the union but why not say so explicitly?

I feel uneasy about the use of the word "genotyping"? No genotype
calls are made as such. After all we are talking about sub-popluation
structure. Would "Variant profiling of sub-populations" be more appropriate?

I'm wondering how the workflow would deal with sub-populations that
contain other bacteria (or contaminants), especially when closely
related. Or is this an intrinsic limitation? This might be worth mentioning. Otherwise users might assume this could be used for e.g. metagenomic data.

The authors use a lot of figures. Some could be moved to the Supplementary information.

Did the authors evaluate the use of the intersection of predictions as well by any chance? That should give a direct boost in specificity. Some users might want to high specificity variants (few FP) instead of high sensitivity mode (exploratory mode), so ideally this could be a user choice (it might require re-evaluating downstream filters though).


Minor comments:
===============

* General question

Despite its ability to predict indels LoFreq was not used for this
because special pre-processing is required?

* Source-code (https://github.com/ma2o/VarCap)

The README is not prominently visible. It could be linked to README.md so
that it gets rendered by default by github.

There is no mention of dependencies or requirements (qsub?).

The necessary configuration of parameters (variant.config) is also not
described

* p8, results, L30:

"At a low MRA of 1% LoFreq-Starshows less sensitivity than
Varscan2. This is to be expected, as LoFreq-Star builds its own error
model and detection threshold to avoid FP and therefore detects no
variants below that threshold"

LoFreq-Star has no hard allele frequency filter, instead it's a
function of coverage and quality. The Varscan2 sensitivity is
surprising. Did the authors forget to run the recommended 'varscan filter'?

* Fig 6

It's unclear what was actually used to produce this plot. For a start:
Which dataset was used? Are these numbers from prefiltered Varcap
results? Why use FP per Megabase instead of FP rate?

* Filter question

Do final variants have to pass MRA *and* MAA ?

* Fig 4 and Fig 5

Why are results there only for 1 and 2% MRA?


Typos etc.:
===========

* Abstract, L8: "including low frequent alleles".

"low frequency" instead?

* Abstract, L13: "that the best sensitivity could be reached by a combination of different tools."

Combination is ambigious as it could refer to intersection or union

* p3, intro, L30: "therefore it is necessary to"

"One way to deal with this is to"

* p4, intro, L11: "All those approaches have been designed for different degrees of heterozygosity"

"zygosity" instead of "heterozygosity"

* p6, methods, L11: "(bwa-0.7.5a, unpublished, Li and Durbin 2009)"

Please cite the biorxiv paper
https://arxiv.org/abs/1303.3997

* p6, methods, L9:

"First a sliding window with size 10 removed any bases with lower quality than 20 from the 3’ side."

Why sliding window?

* p11, l27

"To avoid FP due to reads mapped to repetitive regions, we mask nearly
identical regions according to the properties described above"

Where "above" (the above section is on sequencing errors)

* p15, L11:

"We observed that for SNPs a MRA of 2% was sufficient (MAA of 8 reads)"

The MAA number is only true for a specific coverage setting, right?

Reviewer 2 ·

Basic reporting

-

Experimental design

-

Validity of the findings

-

Additional comments

In "Genotyping of evolving prokaryotic populations" Zojer et al. describe a tool for identifying many different kinds of variants such as SNPs, indels, and structural variants from Pool-Seq data. I think the tool may be useful to some researchers and the manuscript is understandable. However, I have some major and some minor issues:


Major comments:
- it would be good if authors would use more established terms throughout the manuscript; this would enhance understanding of the text and allow for more easy cross-comparision of the results to other works; Sequenced populations of prokaryotes are actually pools, so the authors should use the well established term Pool-Seq; Also I would replace the term MRA (minimum relative abundance) with the more widely used MAF (minor allele frequency)
- also related to the previous comment, I think the authors use the term MRA (minor reference allele) for two entirely different concepts. First for the minor allele frequency (ie. the actual frequency of an allele in the population) and second for the bionformatics threshold used for identifying a SNP (ie the MRA cutoff used by the software). When reading the manuscript I was sometimes unclear whether the authors mean the frequency of a SNP in the population or the cutoff they used for the software. I think understanding of the ms would benefit from distinguishing these two concepts (i.e. use different terms)
- SNP detection has been done with varscan, samtools, lofreq2 and gatk; to my understanding samtools does not work with pools, it was designed for SNP calling in sequenced individuals. Also I do not understand why they did not evaluate more established SNP callers for Pool-Seq. Recent work suggests that for example FreeBayes has an excellent performance with Pool-Seq data. For an overview of SNP callers suitable for Pool-Seq data see http://www.nature.com/nrg/journal/v15/n11/full/nrg3803.html
Actually I think it would be a good idea to evaluate the performance of FreeBayes instead of samtools
- The Breseq pipeline from the Barrick group is doing something very similar as this pipeline; The authors should thoroughly compare the performance and the features of these two pipelines http://barricklab.org/twiki/bin/view/Lab/ToolsBacterialGenomeResequencing
- they write that GATK is only designed for heterozygous or homozygous individuals; this is not true, it can also be used for Pool-Seq but in this case the pool- size needs to be provided. This makes me wonder, did the authors use GATK correctly? Did they provide the pool-size? Which pool-size was provided?
- Discussion: 25-27, they present some rules of how to identify variants, if variants fullfill two of the following three rules; first the authors present actually four rules (not three); second this is not really evaluated; if they present some rules they should evaluate them, showing for example the true positives and false positives for many different rules and than pick the best performing rule


Minor:
- methods 21-23: please rephrase, this procedure is very hard to understand (if at all)
- Results: they report that high confidence intervals need to be supported by two different tools. interestingly there was a recent work which reported the same finding for Pool-Seq data; the authors should cite this work http://www.g3journal.org/content/early/2016/09/09/g3.116.034488.long
- in 3.2.1 they write that it is necessary to use both MRA and MAA and in 3.3.1 they write its only necessary to use MRA; so I'm confused, is it better to use both or to just use MRA?
- 3.3.2 more distantly evolved populations carry higher variant frequencies; I assume because the authors expect abundant positive selection, but it would be good to mention this in the text.
- the authors implemented this tool in galaxy, which is a good idea and may be helpful; However, data upload and processing may be quite slow. So I think the authors should provide time estimates, how long data processing will take for data of different sizes. (maybe separated for different processing steps, like upload, mapping, snpcalling, indels, SVs, download)

---

## Round 0.2 · accepted · Accept

The substantive concerns of the reviewers have all been satisfactorily addressed. There only remain some very minor formatting issues and typos that we recommend be fixed before publication. On p2 l17 "archived" -> "achieved". Starting on p8 inconsistent capitalization of "FreeBayes" vs "Freebayes". On p5 l22 strike "Thereby". Please also see the minor suggestions from the reviewer.

Reviewer 1 ·

Basic reporting

I think my previous comments have all been addressed.

I do have a couple of pedantic suggestions though:
- Please go through the document with a spellchecker to fix typos like 'Magabase' and 'uncomplete'.
- Please check and unify spelling of program names. Most have different case throughout the manuscript, e.g. pindel, Pindel or gatk and GATK, see esp. Fig 2 (along the same lines: it should be "Pool-seq")

Experimental design

no comment

Validity of the findings

no comment